# Effect of Oxaliplatin, Olaparib and LY294002 in Combination on Triple-Negative Breast Cancer Cells

**DOI:** 10.3390/ijms22042056

**Published:** 2021-02-19

**Authors:** Kitti Andreidesz, Balazs Koszegi, Dominika Kovacs, Viola Bagone Vantus, Ferenc Gallyas, Krisztina Kovacs

**Affiliations:** 1Department of Biochemistry and Medical Chemistry, University of Pécs Medical School, 7624 Pécs, Hungary; andreidesz.kitti@pte.hu (K.A.); balazs.koszegi@aok.pte.hu (B.K.); dominika.kovacs@aok.pte.hu (D.K.); viola.vantus@aok.pte.hu (V.B.V.); ferenc.gallyas@aok.pte.hu (F.G.); 2Szentagothai Research Centre, University of Pécs, 7624 Pécs, Hungary; 3Nuclear-Mitochondrial Interactions Research Group, Hungarian Academy of Sciences, 1052 Budapest, Hungary

**Keywords:** olaparib, oxaliplatin, Akt pathway inhibitor, TNBC, MCF7

## Abstract

Triple-negative breast cancer (TNBC) has a poor prognosis as the therapy has several limitations, most importantly, treatment resistance. In this study we examined the different responses of triple-negative breast cancer line MDA-MB-231 and hormone receptor-positive breast cancer line MCF7 to a combined treatment including olaparib, a poly-(ADP ribose) polymerase (PARP) inhibitor, oxaliplatin, a third-generation platinum compound and LY294002, an Akt pathway inhibitor. We applied the drugs in a single, therapeutically relevant concentration individually and in all possible combinations, and we assessed the viability, type of cell death, reactive oxygen species production, cell-cycle phases, colony formation and invasive growth. In agreement with the literature, the MDA-MB-231 cells were more treatment resistant than the MCF7 cells. However, and in contrast with the findings of others, we detected no synergistic effect between olaparib and oxaliplatin, and we found that the Akt pathway inhibitor augmented the cytostatic properties of the platinum compound and/or prevented the cytoprotective effects of PARP inhibition. Our results suggest that, at therapeutically relevant concentrations, the cytotoxicity of the platinum compound dominated over that of the PARP inhibitor and the PI3K inhibitor, even though a regression-based model could have indicated an overall synergy at lower and/or higher concentrations.

## 1. Introduction

In terms of incidence, breast cancer is the leading cancer type among women [1]. It is a heterogenous and hormone-dependent disease [2]; approximately 65–75% of cases are hormone receptor-positive (HR+; estrogen receptor-positive or progesterone receptor-positive) [3], while 15–20% are human epidermal growth-factor receptor 2 (HER2)-positive [4]. The triple-negative subtype (TNBC) represents 15% of all cases [5]. For HR+ and/or HER2+ breast cancers, targeted therapies are available. These include biological and/or hormonal therapy, in which the overexpressed or overactivated molecules are blocked specifically; in the case of HER2+ breast cancer, Herceptin is a widely used monoclonal antibody which binds to HER2, blocking its downstream signalling; in hormone receptor-positive breast cancer, tamoxifen is a regularly used selective estrogen receptor modulator of anti-oestrogenic effects [6]. In contrast, the treatment protocol for TNBC is mainly limited to chemotherapy. Besides the limited therapy options, recurrent tumor resistance and poor prognoses are emerging challenges [7] in TNBC.

The agents used in chemotherapy are rather non-selective as it has been demonstrated by platinum-based substances, which have been in clinical use for decades [8]. Their cytotoxicity is based on the formation of platinum-DNA adducts, leading to double-strand DNA breaks and eventually cell death [8,9]. Cisplatin and carboplatin are widely used platinum-based agents in the treatment of non-small-cell lung cancer, and breast, ovarian and testicular cancer [10]. One major limiting factor in their therapeutic use is the possibility that the cancer cells develop intrinsic or acquired resistance to the treatment [8,11]. Oxaliplatin is a third-generation platinum compound. One of its benefits is decreased mutagenic activity compared to cisplatin and carboplatin, and it is often effective in cisplatin-resistant tumors [8,9]. It has been approved by the FDA for treatment of colorectal cancer [12], and it has the potential to replace other platinum compounds in therapy for other types of cancer including TNBC.

The nuclear Poly(ADP-ribose) polymerase (PARP) enzymes found in all nucleated cells are a family of enzymes involved mainly in DNA repair, namely in base excision and nucleotide excision repair. PARP-1 catalyses transfer of ADP-ribose of NAD^+^ to a wide range of proteins. It binds to specific DNA sequences, and by self-ADP-ribosylation, it recruits repair machinery to the site of DNA damage [13]. PARP inhibition is cytotoxic in germline mutated BRCA1 and BRCA2 ovarian and breast cancer [14]. Olaparib is the most widely used PARP inhibitor in clinical therapy nowadays. It has been approved for the treatment of BRCA 1/2 mutated ovarian and metastatic breast cancer [15]. 

The PI3K/Akt/mTOR pathway play an important role in carcinogenesis and apoptosis resistance, promoting tumor growth and proliferation, and the activation of phosphatidyl-inositol 3-kinase (PI3K) has been associated with endocrine resistance, a major problem in the treatment of breast cancer [16,17,18]. Recently, based on the outcome of the phase III SOLAR-1 clinical trial (NCT02437318), the Food and Drug Administration approved (in combination with fulvestrant, an estrogen receptor antagonist) the PI3K p110α-isoform-specific inhibitor, alpelisib for use in the treatment of HR+/HER2- metastatic breast cancer. However, combination of isoform-specific PI3K inhibitors with immune checkpoint-, receptor- or enzyme inhibitors, including PARP inhibitors, could extend their therapeutic use to HER2+ or triple-negative breast cancers [19]. 

Combination chemotherapy was introduced more than 40 years ago [20]. Many DNA adduct-forming metallodrugs, in combination with other drugs of different molecular action, such as inhibition of protein synthesis or DNA repair mechanisms, can increase their therapeutic efficacy over monotherapy and can overcome chemotherapy resistance [21]. Several preclinical studies found a synergistic effect between platinum drugs and PARP inhibitors. However, experimental design, data acquisition and interpretation of preclinical and especially clinical studies are prone to errors and pitfalls which undermine certainty of this synergy [22]. Even the protective effect of PARP inhibitors against oxidative stress may impede this synergy, since oxidative stress contributes to the therapeutic effect of platinum drugs [23].

Accordingly, the present study aimed to investigate the response of TNBC line MDA-MB-231 versus HR+ breast cancer line MCF7 in a combined treatment comprising oxaliplatin, olaparib and LY294002 treatment.

## 2. Results

### 2.1. Effect of Olaparib, Oxaliplatin and Akt Pathway Inhibitor on Cell Viability

An MTT assay was performed to determine the effect of olaparib, oxaliplatin and LY294002 alone and in combination on MDA-MB-231 and MCF7 cells. As shown in Figure 1A,C, olaparib alone did not decrease viability significantly in either cell line. However, oxaliplatin reduced viability of the TNBC line MDA-MB-231 substantially and, more pronouncedly, the estrogen and progesterone receptor-positive non-TNBC line MCF7. Our results show that olaparib and oxaliplatin had neither a synergistic, nor an additive effect at the applied concentrations, since their combination caused about the same extent of cell death as oxaliplatin alone (Figure 1). The MDA-MB-231 cells were resistant to treatment with the Akt pathway inhibitor LY294002 (Figure 1B). In contrast, LY294002 caused about the same extent of cell death in the MCF7 cell line as oxaliplatin did (Figure 1D). Furthermore, LY294002 enhanced the effect of oxaliplatin but not that of olaparib on both MDA-MB-231 and MCF7 cells. Additionally olaparib did not add to the effect of LY294002 and oxaliplatin (Figure 1B,D).

### 2.2. Effect of Olaparib, Oxaliplatin and Akt Pathway Inhibitor on Cell Death Processes

We investigated which type of cell death LY294002, olaparib and oxaliplatin elicited by flow cytometry after double-staining the treated cells with fluorescently labelled Annexin V and propidium iodide, to demonstrate apoptosis and necrosis, respectively. We found that less than 5% of cell death caused by the compounds investigated was necrotic, and most of the dying cells were in their early apoptotic stage under the experimental conditions we used (Figure 2). Olaparib did not induce apoptosis in MDA-MB-231 and MCF7 cells, while oxaliplatin increased early and late apoptosis in both cell lines (Figure 2). The Akt pathway inhibitor LY294002 alone caused about the same level of apoptosis in both cell lines as oxaliplatin did (Figure 2C,D). On the other hand, treatment of the cells with LY294002 together with olaparib, oxaliplatin or their combination did not significantly change the distribution of cells among live, early, and late apoptotic populations in either of the cell lines (Figure 2C,D).

### 2.3. Effect of Olaparib, Oxaliplatin and Akt Pathway Inhibitor on ROS Production

The reactive oxygen species (ROS) production capacity of olaparib, oxaliplatin and Akt pathway inhibitor LY294002 was measured using a carboxy-H2DCFDA assay. This assay measures fluorescence intensity of H2DCFDA, a fluorescent redox dye oxidized by the ROS from its non-fluorescent reduced form. We found marked differences between MDA-MB-231 (Figure 3A,B) and MCF7 (Figure 3C,D) lines among the treatment groups. The PARP inhibitor increased ROS production in MCF7 cells, although the effect did not reach a statistically significt level (Figure 3C). In contrast, the drug did not induce any ROS production in MDA-MB-231 cells (Figure 3A). Oxaliplatin caused significant ROS production in the triple-negative breast cancer cells which was enhanced by olaparib co-treatment, although olaparib’s enhancing effect did not reach a statistically significt level (Figure 3A). The Akt pathway inhibitor did not affect ROS production either alone or in any combination with olaparib and oxaliplatin (Figure 3B). On the other hand, oxaliplatin did not affect ROS production of MCF7 cells and attenuated ROS production induced either by olaparib (Figure 3C), LY294002, or their combination (Figure 3D). However, these negative effects of oxaliplatin did not reach a statistically significant level. The Akt pathway inhibitor alone induced a similar level of ROS production in MCF7 cells as that caused by olaparib (Figure 3D).

### 2.4. Effect of Olaparib, Oxaliplatin and Akt Pathway Inhibitor on the Cell Cycle

Flow cytometry was used to determine which cell cycle phase the cells had reached after treatment with different combinations of olaparib, oxaliplatin and LY294002. The distribution of control MDA-MB-231 cells among G1, S and G2/M phases was 55.57%, 22.8% and 21.63%, respectively, which was not affected by the PARP inhibitor olaparib (Figure 4A,C), the Akt pathway inhibitor LY294002, or their combination (Figure 4B,D). In contrast, oxaliplatin treatment arrested the cells in the S phase of their cycle (Figure 4C) which was not affected by olaparib co-treatment (Figure 4C). However, LY294002 co-treatment attenuated oxaliplatin’s arresting effect, which was further reduced when olaparib was included in the combination treatment (Figure 4D). 

The cell cycle phase distribution of control MCF7 cells was slightly different from that of MDA-MB-231 cells. Approximately 51.26% of control cells were in G1; 28.64% in S and 20.1% were in G2/M phase (Figure 4E,G). As with the MDA-MB-231 line, oxaliplatin arrested MCF7 cells in the S phase of their cycle (Figure 4G) and olaparib co-treatment did not have any further effect on it (Figure 4G). However, the PARP inhibitor increased the number of G2/M phase cells at the expense of S phase, although this effect did not reach a statistically significant level (Figure 4G). Additionally, LY294002 significantly enhanced the number of G1-phase cells at the expense of S and G2/M phase cells. This effect was not affected by olaparib co-treatment but was significantly counteracted by oxaliplatin (Figure 4H). 

### 2.5. Effect of Olaparib, Oxaliplatin and Akt Pathway Inhibitor on Colony Formation

To assess cellular proliferation capacity, a colony formation assay was performed. Olaparib and oxaliplatin significantly decreased colony numbers of both cell lines, although the non-TNBC line MCF7 was more sensitive to the cytostatic drugs than the TNBC line MDA-MB-231 (Figure 5A,C). Furthermore, oxaliplatin attenuated colony formation to a much greater extent than olaparib did (Figure 5A,C). A combination of olaparib and oxaliplatin caused a similar decrease in colony formationt as oxaliplatin did alone (Figure 5A,C), indicating a lack of synergy, even additivity, between the two substances. LY294002 alone decreased colony formation to about the same extent as olaparib did (Figure 5B,D). The PARP inhibitor andto a greater extent, oxaliplatin both augmented the Akt pathway inhibitor’s effect on colony formation (Figure 5B,D). Again, combination of olaparib and oxaliplatin had about the same effect as the latter alone (Figure 5B,D).

### 2.6. Effect of Olaparib and Oxaliplatin on Invasive Growth

Invasive growth of the cell lines was assessed the xCelligence Real-Time Cell Analysis (RTCA) system. As with the colony formation experiments, olaparib and, to a much greater extent oxaliplatin decreased invasive growth in both cell lines (Figure 6). Again, MCF7 line was more sensitive to the cytostatic drugs than MDA-MB-231, and combination of the two drugs had about the same effect as oxaliplatin alone had (Figure 6). 

### 2.7. Effect of Olaparib, Oxaliplatin and Akt Pathway Inhibitor on Invasive Growth

The Akt pathway inhibitor decreased invasive growth, and its effect on MCF7 cells was more pronounced than on MDA-MB-231 cells (Figure 7). Both olaparib and oxaliplatin treatment augmented LY294002′s effect, decreasing invasive growth to the detection limit (Figure 7).

## 3. Discussion

In breast cancer, TNBC is associated with poorer prognosis and limited targeted therapeutic options compared to the HR+ subtype [6]. Accordingly, in this study, we investigated the response of different types of breast-cancer cell lines to a combination of conventional chemotherapy [24] and synthetic lethality-based therapy [25] supplemented with Akt inhibition to prevent undesired cytoprotective effects of the latter [26]. We used the TNBC cell line MDA-MB-231 and estrogen and progesterone receptor-positive cell line MCF7, and treated these with the third-generation platinum compound oxaliplatin, the PARP inhibitor olaparib and the PI3K inhibitor LY294002.

Several studies have reported the synergistic effect of PARP inhibitors and chemotherapeutic platinum agents in various tumors. Olaparib and platinum compound carboplatin have been found to have modest activity in patients with sporadic TNBC [27]. Combination of PARP inhibitor PJ34 and antineoplastic agent cisplatin has been found to have cytotoxic synergy in non-small-cell lung-cancer line A549 [28]. Furthermore, PJ34 enhanced suppressive effect of cisplatin in liver-cancer cell line HepG2 [29]. The importance of the PI3K/Akt pathway in therapy resistance has been highlighted, demonstrating that its activation results in decreased sensitivity to chemotherapeutic agents [30]. Furthermore, PI3K/Akt pathway inhibitors have been known to cause more favorable outcomes when co-administered with usual anticancer drugs [31]. To provide experimental support for the rationale of combination therapy in TNBC, Zhao et al. investigated various combinations of olaparib, carboplatin and buparlisib, a pan-PI3K inhibitor in two human TNBC lines and a HR+ breast-cancer line [32]. By using a calculation [22] based on the median-effect equation, they found a synergistic cytostatic effect of the combination therapy in TNBC lines but not in the HR+ line [32]. We approached the question of synergy from a practical point of view. Instead of determining dose-response effects, we used single, therapeutically relevant concentration of each drug, and applied these individually and in all possible combinations in experiments on viability, type of cell death, ROS production, cell-cycle phase, colony formation and invasive growth.

We found that 72 h of olaparib treatment decreased viability of MCF7 cells to a much greater extent than that of MDA-MB-231 cells (Figure 1). Elevated PARP-1 expressions have been reported in a wide range of human cancers including breast cancer, and an especially high PARP-1 expression has been found found in TNBC which can explain our results [23]. Furthermore, in complete agreement with our data,other studies have found the cytotoxic effect of oxaliplatin to be higher in MCF7 than in TNBC cells [30]. Several studies have reported the synergistic cytostatic effect of PARP inhibitors and platinum agents [27,28,29], and one study reported synergism in combined therapy comprising olaparib, carboplatin and the PI3K inhibitor buparlisib in TNBC lines but not in a HR+ breast-cancer line [32]. In contrast, under our experimental conditions, olaparib did not enhance the cytotoxic properties of oxaliplatin (Figure 1), and we could not detect synergism, nor even an additive effect between these two drugs. The PI3K inhibitor LY294002 decreased viability of the TNBC but not the HR+ line when combined with olaparib, oxaliplatin or both. However, these effects did not reach a statistically significant level (Figure 1). These data compellingly indicate that, at therapeutically relevant concentrations, cytotoxicity of the platinum compound dominated that of the PARP inhibitor and the PI3K inhibitor. At lower platinum compound and higher concentrations of PARP and PI3K a synergistic effect likely appear and a regression-based model could indicate an overall synergy that may explain the conflict between our results (Figure 1) and the findings of others [27,28,29,32]. Additionally, platinum compounds induce ROS production [33] and PARP inhibitors are known to protect against oxidative stress [23], what could contribute to the absence of synergy between the PARP inhibitor and the platinum agent that we observed. Accordingly, blocking the PI3K/Akt pathway by the PI3K inhibitor LY294002 increased the cytotoxicity of olaparib and oxaliplatin co-treatment, although the effect did not reach a statistically significant level (Figure 1). 

We found that olaparib and oxaliplatin killed MDA-MB-231 and MCF7 cells predominantly by apoptosis (Figure 2). The apoptosis resistance of the two cell lines is different. MDA-MB-231 line has high levels of mutant p53 [34], whereas MCF7 line has wild-type p53 [35]. Additionally, TNBC cells have 10-fold greater phospholipase D (PLD) activity than MCF7 cells [34]. Mutant p53 and elevated PLD activity play a significant role in the survival of cancer cells and can contribute to the suppression of apoptosis [34]. Nevertheless, the effect of the various treatments on distribution among live, early and late-apoptotic populations was similar in both cell lines (Figure 2). In this respect it is worth noting that the washing steps before and after the staining procedure remove most non-adherent cells, and the flow cytometry method used to determine the type of cell death analyses stained cells only, regardless of the original cell number and their sensitivity to the various treatments. 

Among other mechanisms, ROS-mediated processes play a prominent role in remodelling cancer phenotypes resistant to apoptosis which acquire enhanced metastatic properties [36]. In solid tumors, hypoxia and the resulting hypoxia-inducible factor (Hif)-1α mediated metabolic plasticity play a pivotal role in malignant transformation [37]. However, in cell culturing conditions, uniform oxygen partial pressure and practically inexhaustible extracellular fuel supply obscure these processes. Accordingly, we studied ROS production which reflects metabolic plasticity [38] and is compatible with cell culturing conditions. Increased ROS production by the platinum compounds [33] could induce DNA breaks that may accumulate when PARP is inhibited leading to cell death. Such a mechanism could account for the observed synergism between platinum compounds and PARP inhibitors [27,28,29]. In complete agreement with the literature, we found that oxaliplatin- but not olaparib or the Akt pathway inhibitor LY294002- induced ROS formation in the TNBC MDA-MB-231 line and LY294002, but olaparib did not augment oxaliplatin’s effect (Figure 3). On the other hand, the treatments alone or in combination failed to induce significant ROS production in the non-TNBC MCF7 line (Figure 3). However, increased vulnerability of MCF7 cells to the treatments resulted in a death rate, compared to that of MDA-MB-231 cells (Figure 1), leaving fewer surviving cells to produce ROS. Furthermore, MCF7 cells could produce less ROS as they represent an earlier stage of metabolic transformation than the TNBC MDA-MB-231 cell line does. Combination of these and possibly other factors could account for the observed difference between the cell lines. 

Centrosome amplification occurring in the S phase of the cell cycle, is known to be associated with malignant transformation in various tissue types. Centrosome amplification is regarded as a marker for aggressiveness, even with invasive breast and prostate cancers [39]. Accordingly, we expected to find that the MDA-MB-231line had a higher percentage of cells in the S phase of their cycle than the MCF7 cells. However, we observed the opposite trend in the two cell lines (Figure 4C vs. Figure 4G) indicating that other factors, probably synchronization of cell cycles due to passages during culturing [40] dominated centrosome amplification in determining the distribution of cell-cycle phases in these two breast cancer cell lines. In both cell lines, although to a different extents, oxaliplatin arrested most of the cells in their S phase, and this was not affected by olaparib co-treatment (Figure 4C,G). These data are consistent with the DNA crosslinking effect of the platinum compound, which prevents cells from crossing the G2 checkpoint.

The TNBC cell line MDA-MB-231 represents a more aggressive, apoptosis- and therapy-resistant phenotype than the non-TNBC MCF7 line does. As measures of this aggressiveness, we assessed colony formation (Figure 5) and invasive growth (Figure 6 and Figure 7). These data were completely consistent with the results for viability (Figure 1), with the literature and with the aforementioned view about aggressiveness of the two cell lines. They provide two additional experimental evidence for the lack of synergy between olaparib and oxaliplatin (Figure 5, Figure 6 and Figure 7). Furthermore, they indicate (Figure 5B,D) that Akt pathway inhibition could be advantageous in combined therapy with PARP inhibitors, as it blocks their Akt-mediated cytoprotective effects [26].

In conclusion, we provided experimental evidence for the lack of synergy between olaparib, a PARP inhibitor widely used in cancer therapy, and oxaliplatin, a third-generation platinum compound. These results are in conflict with the findings of others [27,28,29,32], probably because, at therapeutically realistic concentrations, the cytostatic effect of the platinum compound dominates that of the PARP inhibitor. We have also demonstrated the advantage of using an Akt pathway inhibitor to augment the cytostatic properties of the platinum compound and/or to prevent the cytoprotective effects of PARP inhibition. Furthermore, we have shown the therapy resistance of the TNBC line MDA-MB-231 over the estrogen- and progesterone receptor-positive line MCF7, although we failed to advance our understanding of differences in sensitivity to chemotherapy among different types of breast cancers.

## 4. Materials and Methods

### 4.1. Drugs

Olaparib was purchased from MedChemExpress (Monmouth Junction, NJ, United States). It was dissolved in DMSO before treatment and for each treatment new solution was made. Oxaliplatin was purchased from Accord Healthcare (Munich, Germany). Akt pathway inhibitor LY294002 was purchased from Selleckchem (Houston, Texas, USA). All other reagents were of the highest purity commercially available. 

### 4.2. Cell Cultures

MDA-MB-231 and MCF7 cell lines were obtained from American Type Culture Collection (Manassas, VA, United States). Cell lines were maintained in a humidified 5% CO_2_ atmosphere at 37 °C. MDA-MB-231 cells were cultured in DMEM low glucose (Biosera, Nuaille, France) supplemented with 10% (*v/v*) FBS (Thermo Fisher, Life Technologies, Milan, Italy). MCF7 cells were cultured in RPMI (Biosera, Nuaille, France) supplemented with 10% (*v/v*) FBS. 

### 4.3. Survival Assay

MTT assay was used to examine cell viability. Cells were seeded at a density of 3 × 10^3^/well in 96-well cell culture plates for 24 h before treatment. After 72 h treatment using olaparib (2 µM), oxaliplatin (25 µM) and LY294002 (1 µM), and their combination, the medium was replaced containing 0.5% MTT substrate (100 µL/well). After 2 h incubation at 37 °C the medium was discarded and DMSO was added (100 µL/well). Given the purple color of the soluble formazan product, absorbance was measured at 570 nm using the GloMax^®^-Multi Instrument (Promega, Madison, WI, United States). At least four parallels were used, and the experiment was repeated three times.

### 4.4. Apoptosis Assay

Live, early apoptotic, late apoptotic and dead cells were quantified using the MUSE Annexin V and Dead Cell Kit (Luminex Corporation, Austin, Texas, United States). The experiment was carried out according to the manufacturer’s instructions. Cells were cultured in six-well plates (1.5 × 10^5^/well) and treated with olaparib (2 µM), oxaliplatin (25 µM) and LY294002 (1 µM) and their combination for 72 h. After the treatment cells were collected and diluted in their medium to a concentration of 5 × 10^5^/mL. Twenty minutes incubation in the dark at room temperature followed the addition of 100 µL Annexin V reagent. Annexin V-FITC positive cells were considered as apoptotic (early and late apoptotic) and were expressed as % of the total cell number examined. The MUSE Cell Analyzer device was used to assess the samples and 5000 single cell events were measured per sample. Three independent experiments were performed.

### 4.5. Assay for Reactive Oxygen Species

To measure intracellular reactive oxygen species (ROS) production, cells were plated (3 × 10^3^/well) in wells of a 96-well plate, were cultured for 24 h were treated for 72 h with olaparib (2 µM), oxaliplatin (25 µM) and LY294002 (1 µM), and their combination. The medium was replaced by Krebs-Henseleit solution containing 2% FBS and 2 µM carboxy-H2DCFDA (Thermo scientific, Waltham, MA, United States). After 30 min incubation, ROS generation was measured using GloMax^®^-Multi Instrument (Promega, Madison, WI, United States) at respective excitation/emission wavelengths of 490/530 nm.

### 4.6. Cell Cycle Analysis

Cells were cultured in six-well plates (1.5 × 10^5^/well) 24 h before treatment, and were treated with olaparib (2 µM), oxaliplatin (25 µM) and LY294002 (1 µM), and their combination, for 72 h. To assess cell cycle, flow cytometry analysis was applied. In brief, cells were harvested, washed with Dulbecco’s phosphate-buffered saline (DPBS) (Biosera, Nuaille, France) and fixed with 70% ethanol (Molar Chemicals Kft., Halasztelek, Hungary) at 4 °C overnight. After fixation, the cells were centrifuged and washed twice with DPBS. Aliquots of cells (0.5 × 10^6^) were stained with 500 µl propidium iodide (PI) (Merck KGaA, Darmstadt, Germany)/RNAse A (Thermo scientific, Waltham, MA, United States) solution containing 0.02 mg/mL PI and 10 µg/mL RNAse A in PBS-0.1% Triton-X 100 (Sigma, St. Louis, MO, United States). SONY SH800 Cell Sorter (SONY Biotechnology, San Jose, CA, United States) was used to measure fluorescent intensities. Debris and aggregates had been discriminated by gating, and at least 5000 single cell events were measured per sample. Data were analyzed and cell-cycle distribution were determined using ModFit LT (Verity software House, Topsham, ME, United States) software.

### 4.7. Clonogenic Assay

Cells were seeded at adensity of 3 × 10^3^/well and were cultured for 24 h before treatment. After 14 days of treatment with 2 µM olaparib, 25 µM oxaliplatin and 1 µM LY294002, alone and in combination, the cells were washed with 1× PBS (Biowest, Nuaille, France) and stained with 0.1% Coomassie Brilliant Blue R 250 (Merck KGaA, Darmstadt, Germany) in 30% methanol and 10% acetic acid. Plates were scanned and colonies were quantified using ImageJ program.

### 4.8. Growth Measurement

Cells were seeded at a density of 1 × 10^3^/well and cultured for 24 h before treatment with 2 µM olaparib, 25 µM oxaliplatin and 1 µM LY294002, alone and in combination, in an electronic microtiter plate (E-Plate^®^) (ACEA Biosciences, San Diego, CA, United States). The treatment lasted 7 days. The impedance was measured every hour, using gold electrodes at the bottom of the wells. The xCELLigence Real-Time Cell Analysis (RTCA) device (ACEA Biosciences, San Diego, CA, United States) was used according to the manufacturer’s protocol. The instrument was placed in a humidified incubator at 37 °C and 5% CO_2_.

### 4.9. Statistical Analysis

Data were analyzed using OriginPro^®^ software. For determining differences among the treatment groups, one-way ANOVA with Tukey post hoc comparison tests were performed on the row data (never on the calculated data). Results are presented as mean ± SEM of at least 3 independent experiments, as indicated in the figure legends. The differences among the groups were regarded as significant at *p* < 0.05.

## Figures and Tables

**Figure 1 ijms-22-02056-f001:**
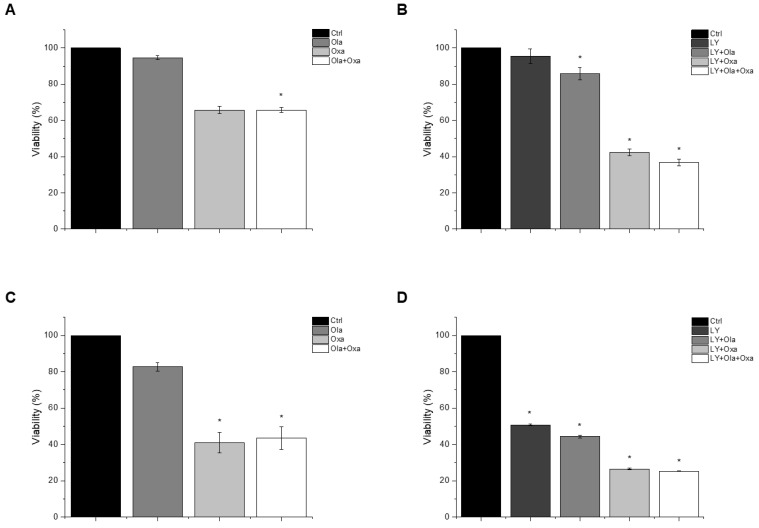
The effects of Ola, Oxa and LY on the viability of TNBC MDA-MB-231 (**A**,**B**) and non-TNBC MCF7 (**C**,**D**) cells were evaluated by MTT assay. Cells were treated with 2 µM olaparib, 25 µM oxaliplatin, 1 µM LY294002 alone and in combination for 72 h. Data are shown as mean ± SEM of at least three separate experiments. * *p* < 0.05 compared with the untreated cells. Ola—olaparib; Oxa—oxaliplatin; LY—LY294002; TNBC—triple-negative breast cancer; MTT-3-(4,5-dimethylthiazol-2-yl)-2,5-diphenyltetrazolium bromide.

**Figure 2 ijms-22-02056-f002:**
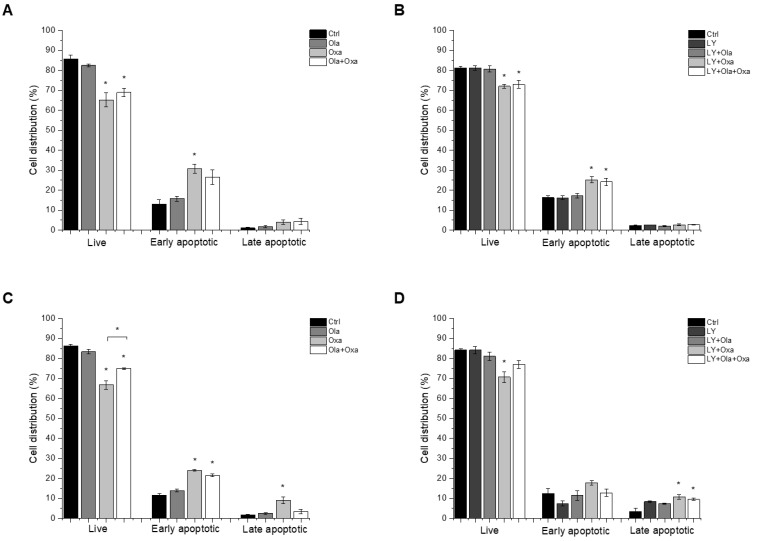
The effects of Ola, Oxa and LY on MDA-MB-231 (**A**,**B**) and MCF7 (**C**,**D**) cell apoptosis were detected by MUSE Cell Analyzer using MUSE Annexin V and Dead Cell Kit after 72 h treatment with 2 µM olaparib, 25 µM oxaliplatin, 1 µM LY294002 alone and in combination. Results are shown as mean ±SEM of at least three separate experiments. * *p* < 0.05 compared with the untreated cells. Ola—olaparib; Oxa—oxaliplatin; LY—LY294002.

**Figure 3 ijms-22-02056-f003:**
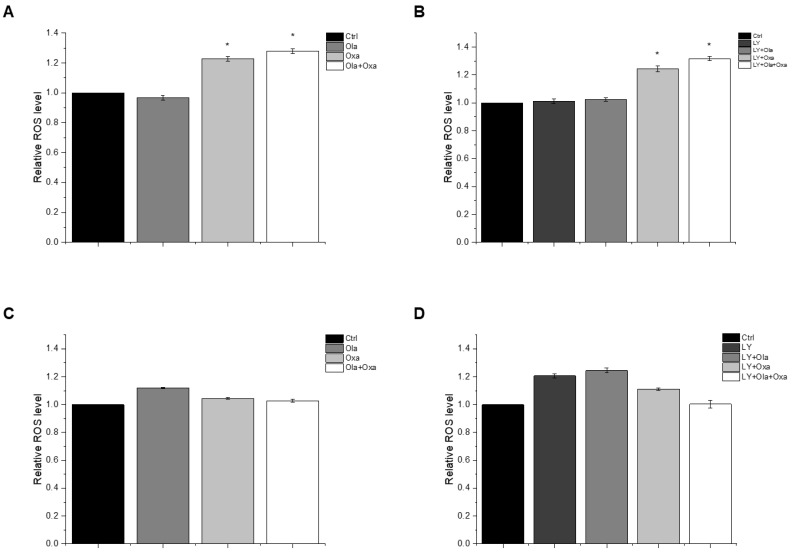
Effects of Ola, Oxa and LY on ROS production by MDA-MB-231 (**A**,**B**) and MCF7 (**C**,**D**) cells treated with 2 µM olaparib, 25 µM oxaliplatin and 1 µM LY294002. Results are shown as mean ±SEM of at least three separate experiments. * *p* < 0.05 compared with the untreated cells. Ola—olaparib; Oxa—oxaliplatin; LY—LY294002.

**Figure 4 ijms-22-02056-f004:**
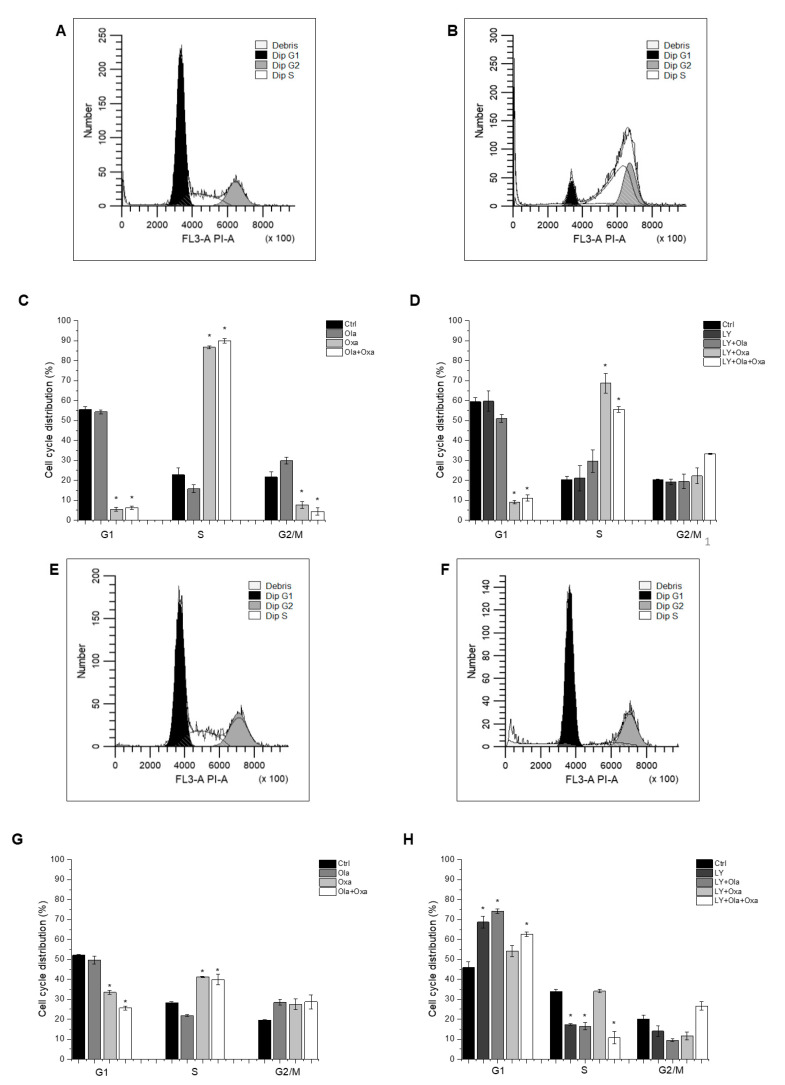
Flow cytometry analysis of MDA-MB-231 (**A**–**D**) and MCF7 (**E**–**H**) cell-cycle distribution. Cells were cultured and treated for 72 h with 2 µM olaparib, 25 µM oxaliplatin and 1 µM LY294002 alone and in combination. The cell-cycle distribution was determined with propidium iodide staining. The histograms show cell-cycle phases of control cells (**A**,**E**) and cells treated with combination of olaparib, oxaliplatin and LY294002 (**B**,**F**). The bar charts represent effect of single and combined treatment on cell cycle phase distribution (**C**,**D**,**G**,**H**). Results are expressed as mean ±SEM of at least three separate experiments. * *p* < 0.05 compared with the untreated cells.

**Figure 5 ijms-22-02056-f005:**
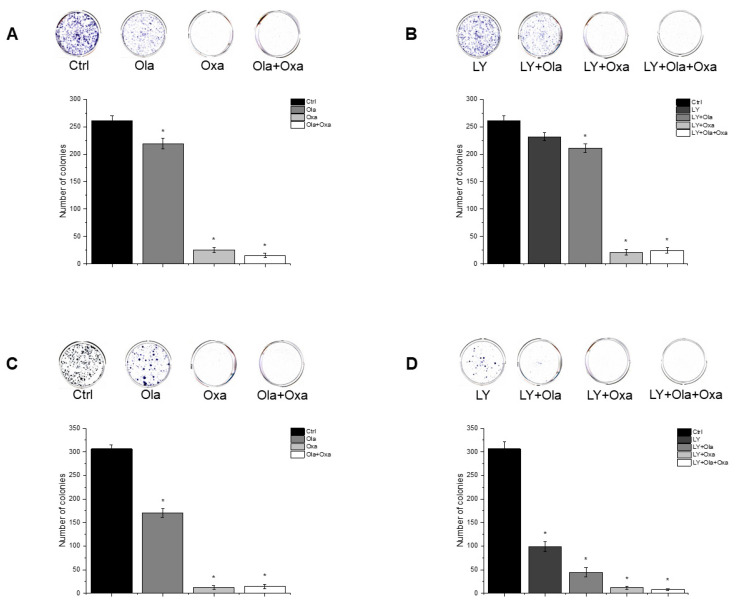
Effects of Ola, Oxa and LY on colony formation by MDA-MB-231 (**A**,**B**) and MCF7 (**C**,**D**) cells. Cells were seeded in six-well plates and after one day treated with 2 µM olaparib, 25 µM oxaliplatin, 1 µM LY294002 alone and in combination. Cells were treated for 14 days before being stained with Coomassie Blue. Results are shown as mean ± SEM of at least three separate experiments. * *p* < 0.05. Ola—olaparib; Oxa—oxaliplatin; LY—LY294002.

**Figure 6 ijms-22-02056-f006:**
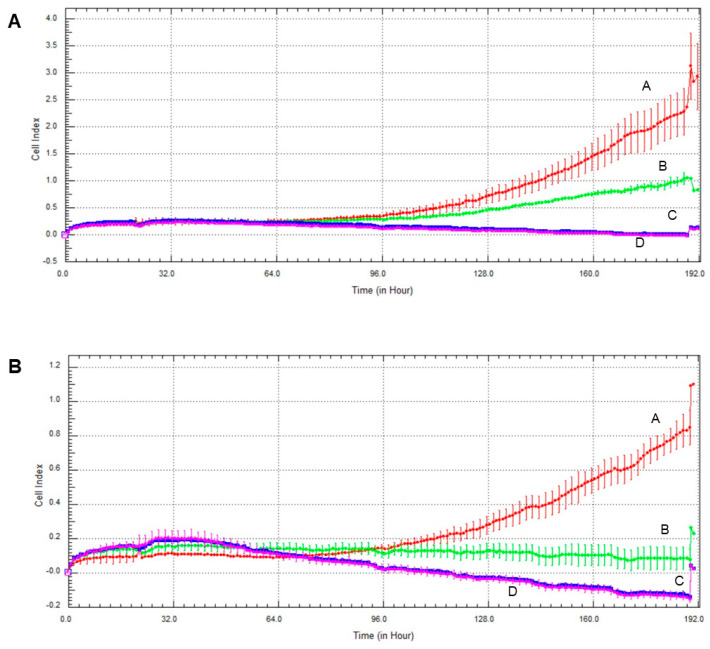
Effect of olaparib and oxaliplatin on invasive growth of MDA-MB-231 (**A**) and MCF7 (**B**) cells treated with 2 µM olaparib and 25 µM oxaliplatin. Line A—control, Line B—2 µM olaparib treatment, Line C—25 µM oxaliplatin treatment, Line D—combination treatment.

**Figure 7 ijms-22-02056-f007:**
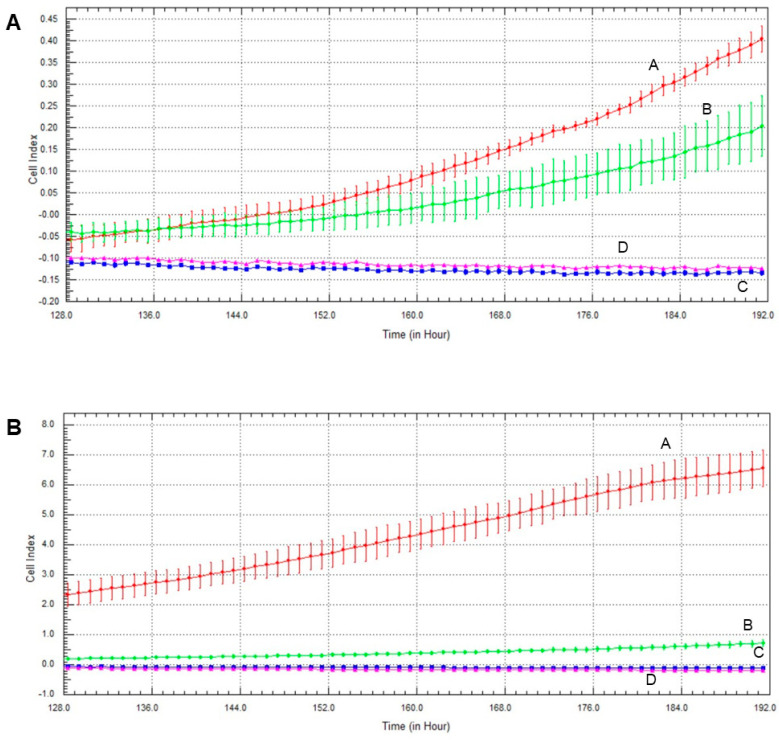
Effects of olaparib, oxaliplatin and LY294002 on invasive growth of MDA-MB-231 (**A**) and MCF7 (**B**) cells treated with 2 µM olaparib, 25 µM oxaliplatin and 1 µM LY294002. Line A—control, Line B—1 µM LY294002 treatment, Line C—2 µM olaparib and 25 µM oxaliplatin, Line D—combination treatment of 1 µM LY294002, 2 µM olaparib and 25 µM oxaliplatin.

## Data Availability

All data generated or analyzed during this study are included in this published article.

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
