# Peer review of "Effect of Oxaliplatin, Olaparib and LY294002 in Combination on Triple-Negative Breast Cancer Cells"

_ijms, 2021, doi:10.3390/ijms22042056_

Round 1
Reviewer 1 Report
This study investigated the combinative effects of platinum compound (oxaliplatin), PARP inhibitor (olaparib), and Akt inhibitor (LY294002) against TNBC and ER+ breast cancer cells. The authors provide a comprehensive study on a combined therapeutic approach for the treatment of breast cancer. However, the authors need to clarify some points in the current literature.
(1) Figure 4. Can you provide the original data of Figure 4C, D, G, and H, as shown in Figure 4A, B, E, and F?
(2) Many studies have demonstrated the effects of combination treatment with oxaliplatin and olaparib in various cancer types, such as colon cancer and ovarian cancer. The authors might explain what the current findings are similar to and what differs from previously published reports. Also, the authors should describe the novelty of the current results in the discussion section.
Author Response
Reviewer 1
We thank the reviewer his/her thorough work. We believe that his/her contribution has improved the manuscript considerably.
1/ We provided the original data of Figure 4C, D, G, and H as requested by the reviewer.
2/ We compared our data with those provided by other groups. We tried to explain the discrepancy between them, and provided references to support our explanation. We also did our best to highlight the novelty of our results according to the reviewer’s suggestion.
Additionally, we intend to improve the English of the manuscript, but the deadline does not allow enough time for both revision and English editing. Therefore, we send the clear version of the revised manuscript to a commercial Scientific Editing company, and incorporate their correction into the next round of revision or the galley proofing step.

Reviewer 2 Report
The authors have done systematic research on the combinatorial treatment with PAPP inhibitor Olaparib, platinum compound Oxaliplatin and PI3K inhibitor LY294002 on two different breast cancer cell lines MDA-MB-231 and MCF7. The authors have collected a comprehensive dataset based on the experimental design encompassed metrics including cell viability, cell death process, ROS production, cell cycle, colony formation, and invasive growth. However, I have some comments and suggestions for the authors' concern as follows. Hope it can help the authors to improve the quality of this manuscript and enhance the attractiveness to readers.
- The Figure 4 should be modified with higher resolution or split into several separate panels
- The abstract should be re-organized with more clear and precise description of the significance, method, result and conclusion of your research
- The English writing should get improved and I suggest the authors to ask a native English speaker for pre-reading
- I suggest the authors to figure out a better title for this manuscript to fully summarize the central point of your research. For the current title, some parts were missing such as PI3K inhibitor LY294002 and I don’t think the result you obtained was able to support the argument in the title as I can’t find proof that there was any combinative effect (synergistic effect) between Olaparib and Oxaliplatin based on your result.
- If the authors desire to claim the synergistic effect between Olaparib and Oxaliplatin, more comparative analysis need to be conducted between single treatment group with Olaparib or Oxaliplatin and co-treatment group with Olaparib and Oxaliplatin.
- There is a similar research published on Oncol Rep. 2018, 40(6):3223-3234 by Zhao, H. et al. I suggest the authors to read this reference carefully and I am sure you will get inspired from how they analyzed and presented their result.

Author Response
Reviewer 2.
We thank the reviewer his/her thorough work. We believe that his/her contribution has improved the manuscript considerably.
1/ We have modified Fig 4, increased the resolution and increased the number of panels as requested by the reviewer
2/ We revised the abstract according to the suggestions of the reviewer.
3/ We definitely intend to improve the English of the manuscript, as requested by the reviewer. However, the deadline does not allow enough time for both revision and English editing. Therefore, we send the clear version of the revised manuscript to a commercial Scientific Editing company, and incorporate their correction into the next round of revision or the galley proofing step.
4/ We modified the title according to the reviewer’s suggestions.
5/ We are sorry that our intention concerning synergy was not clear enough. In fact, at the therapeutically realistic concentrations we used in our experiments, we found lack of synergy, which finding is in conflict with the reports of others. We did our best to explain this discrepancy, and provided references to support our explanation.
6/ In fact, in all our presented experiments, we treated the cells with each compound individually and with all their possible combinations. However, we used a single, therapeutically realistic concentration for each compound only. In most cases, the platinum compound’s effect dominated over that of the others, therefore, we could not detect synergy, often not even an additive effect during the combination treatments.
7/ We discussed the study by Zhao et al. and compared their findings with ours. We indeed thank the reviewer for this excellent suggestion, which helped us to understand the discrepancy between our findings and those of others. Figure 1 in Zhao’s paper clearly indicates that at therapeutic and lower concentrations cytotoxicity of the drugs affected the TNBC cells in a smaller extent than it did the HR+ cells, which results are completely in line with ours. At higher than therapeutic concentrations, the TNBC cells were more vulnerable than the HR+, which, due to the regression analysis resulted in an overall synergistic effect in Zhao’s study.

Round 2
Reviewer 2 Report
The authors have improved remarkably after revision on both English writing and scientific presentation. In this revised version, the readers will clearly get the point what they have done and what the conclusion they have made based on their result.